# Epigenetic Differences Arise in Endothelial Cells Responding to Cobalt–Chromium

**DOI:** 10.3390/jfb14030127

**Published:** 2023-02-26

**Authors:** Célio Junior da C. Fernandes, Rodrigo A. Foganholi da Silva, Gerson Santos de Almeida, Marcel Rodrigues Ferreira, Paula Bertin de Morais, Fábio Bezerra, Willian F. Zambuzzi

**Affiliations:** 1Lab of Bioassays and Cellular Dynamics, Department of Chemical and Biological Sciences, Institute of Biosciences, UNESP-São Paulo State University, Botucatu 18618-970, SP, Brazil; 2Department of Dentistry, University of Taubaté, Taubaté 12020-340, SP, Brazil; 3Program in Environmental and Experimental Pathology, Paulista University (UNIP), Campus São Paulo, São Paulo 04026-002, SP, Brazil

**Keywords:** angiogenesis, biomaterial, cobalt–chromium, DNA methylation, endothelial cells, epigenetics, histone acetylation

## Abstract

Cobalt–chromium (Co-Cr)-based alloys are emerging with important characteristics for use in dentistry, but the knowledge of epigenetic mechanisms in endothelial cells has barely been achieved. In order to address this issue, we have prepared a previously Co-Cr-enriched medium to further treat endothelial cells (HUVEC) for up to 72 h. Our data show there is important involvement with epigenetic machinery. Based on the data, it is believed that methylation balance in response to Co-Cr is finely modulated by DNMTs (DNA methyltransferases) and TETs (Tet methylcytosine dioxygenases), especially DNMT3B and both TET1 and TET2. Additionally, histone compaction HDAC6 (histone deacetylase 6) seems to develop a significant effect in endothelial cells. The requirement of SIRT1 seems to have a crucial role in this scenario. SIRT1 is associated with a capacity to modulate the expression of HIF-1α in response to hypoxia microenvironments, thus presenting a protective effect. As mentioned previously, cobalt is able to prevent HIF1A degradation and maintain hypoxia-related signaling in eukaryotic cells. Together, our results show, for the first time, a descriptive study reporting the relevance of epigenetic machinery in endothelial cells responding to cobalt–chromium, and it opens new perspectives to better understand their repercussions as prerequisites for driving cell adhesion, cell cycle progression, and angiogenesis surrounding this Co-Cr-based implantable device.

## 1. Introduction

Biomaterials play a key role in the success of bone reconstruction, which, for decades, has been widely used in the fields of implantology, dentistry, and regenerative medicine [1]. Overall, a biomaterial can be described as a systematic material and a pharmacologically inert substance designed for implantation within a living system [2] in order to meet the requirements of regenerative procedures for the regeneration of missing tissues and organs. Among biomaterials, autologous biomaterials are considered the gold standard for treating periodontal defects and bone regeneration [3]. However, considering their limited availability, other options have been proposed and are constantly in development [4,5,6,7].

In this sense, studies have been carried out in order to propose biomaterials that can help or replace tissues in their biological functions [8], as well as to propose surface modifications to biomaterials already known, such as titanium alloys or other widely used materials in clinical procedures, such as cobalt–chromium (Co-Cr) and zirconia. It is expected that those modifications will become more bioactive materials, focusing on more appropriate biological responses from the host tissue; this is because the surface directly interacts with the surrounding tissue by promoting reactional tissue during appositional bone growth [9,10,11,12,13]. Among the surface’s modifications, dual acid etching is known to at least promote irregular topography and increase the contact area with host cells, such as osteoblasts and endothelial cells [14,15,16,17].

Over the last few years, we have demonstrated that metallic biomaterials are able to modulate the specific biological responses of surrounding tissues by driving intracellular signaling that is able to modulate gene expression in cells [18,19,20], mainly considering the cellular mechanisms of cell adhesion, proliferation, and differentiation [21,22,23,24]. Among the metallic materials used in clinical trials, Co-Cr alloys have shown that there is no cytotoxic or carcinogenic effect on responsive cells [18], and they display very interesting mechanical resistance; altogether, these characteristics are extremely attractive for alternative biomaterials to be used in humans [18,25,26,27]. Although the literature already extensively discusses these characteristics, it is barely known how the biological mechanisms triggered by these materials work considering the pleiotropy of the cells composing the surrounding tissue during osseointegration mechanisms. In addition, it is also believed that biomaterials develop antimicrobial activity on their metal surfaces, and this opens new perspectives in the development of inherent antibacterial medical devices [28,29,30,31].

Among the biological mechanisms of the challenged cells, using epigenetic marks seem to be an adequate strategy to better understand the effect of the microenvironment promoted by Co-Cr in affecting the phenotype of responsive cells. We have gained some experience on this topic by applying this strategy to other known biomaterials [32], such as titanium and zirconia [33]. Regarding Co-Cr, we have previously described its effect on fibroblasts and pre-osteoblasts, where the activity of mitogen-activated protein kinase (MAPKs) seems to be determinant in drive-cell fates. It is important to mention that cobalt mimics hypoxia, thereby preventing the degradation of hypoxia-inducible factor 1 subunit alpha (HIF1A); this environmental condition deserves more attention, mainly considering the impact of Co-Cr-enriched mediums in modulating the phenotype of endothelial cells, and it is believed to modify the epigenetic machinery.

Epigenetic machinery involves the control of DNA methylation with a prominent role in transcriptional regulation, specifically regarding the silencing of a specific gene. Conversely, demethylation, which consists of the removal of a methyl group from a DNA nucleotide, may control imprinted gene expressions; at the same time, the most important carrier of epigenetic information is the post-translational modification of histones, which have emerged as one of the prime constituents of transcription regulatory machinery. Among them is the methylation and acetylation of lysine residues, although lysine ubiquitylation and serine/threonine/tyrosine phosphorylation are the main means of epigenetic control [34].

Taking into consideration the relevance of epigenetic mechanisms during cell responses to Co-Cr, we have investigated it by analyzing the expression of genes and protein content in the classical biomarkers of epigenetic pathways, such as DNA methylation (practiced by DNA methyltransferase-DNMT) and demethylation (controlled by Tet methylcytosine dioxygenase-TET) [35,36,37,38]. Conversely, we have also investigated the action of histone acetyltransferase (HAT) and histone deacetylase (HDAC) enzymes in endothelial cells. In summary, our data are a descriptive set of data highlighting the relevance of epigenetic marks in endothelial cells responding to Co-Cr, and this opens new perspectives to propose functional methodologies to better know their effects on endothelial cell viability and angiogenesis.

## 2. Materials and Methods

Materials: Cobalt–chrome (Co-Cr) was obtained from SIN Implants System Company (SIN), São Paulo, Brazil. Antibodies: HDAC1 (10E2) Mouse mAb #5356; HDAC2 (3F3) Mouse mAb #5113; HDAC3 (7G6C5) Mouse mAb #3949; HDAC6 (D2E5) Rabbit mAb #7558; SirT1 (1F3) Mouse mAb #8469; SAPK/JNK antibody (phospho-Thr180/ Tyr182) #4511; GAPDH (D16H11) Rabbit mAb #5174. These were obtained from Cell Signaling (Danvers, MA, USA). DNMT1 antibody (60B1220.1), DNMT3A antibody (64B1446) (Novus Biologicals LLC, Centennial, CO, USA), TET3 antibody, anti-DNMT3B (orb372330), TET1 (orb228563), and TET2 (orb131790) were purchased from BiorByt (San Francisco, CA, USA). Anti-ERK1/ERK2 antibody (ERK-7D8) (ab54230); anti-ERK1/2 (phospho-Thr202/ Tyr204) antibody (ab214362); anti-P38 (ab7952); and anti-P38 (phospho-T180 1 Y182]) (ab4822) were obtained from Abcam (Cambridge, MA, USA).

Cell culture: Human Umbilical Vein Endothelial Cells–HUVEC (ATCC-CRL-1730) was provided by ATCC and used in agreement with its recommendations. Cells were maintained in Dulbecco’s Modified Eagle’s medium (DMEM; Sigma Chemical Co., San Luis, Missouri, USA) supplemented with 10% fetal calf serum (FCS; Gibco, Grand Island, NY, USA), 100 U/mL of penicillin, and 100 μg/mL of streptomycin at 37 °C in a humidified atmosphere containing 5% CO_2_.

Co-Cr-enriched medium: The conditioned medium was prepared according to ISO10993-12:2016 by incubating discs of Co-Cr in a conic tube containing DMEM without SFB for up to 24 h. Thereafter, the Co-Cr-enriched medium was harvested to further expose HUVECs [18].

Cell exposition and experimental design: The HUVEC cells (3.5 × 10^4^/^mL^) were seeded in sextuplicate into 96-well plates in DMEM supplemented with 10% FCS. Thereafter, after 72 h of incubation at 37 °C in a humidified atmosphere containing 5% CO_2_, the cells were exposed to Co-Cr-enriched medium respecting the experimental design as follows: control—the cells were maintained under classical conditions; Co-Cr/Wo—the cells were treated with conditioned medium obtained from Co-Cr without DAE; Co-Cr/W—the cells were treated with conditioned medium obtained from Co-Cr discs subjected to DAE. All the cultures were incubated for 72 h at 37 °C in a humidified atmosphere containing 5% CO_2_.

Western blot: After each treatment, challenged HUVEC cells were properly washed in ice-cold PBS and the protein extracts were obtained using a lysis buffer (50 mM Tris-HCl, pH 7.4, 1% Tween 20, 0.25% sodium deoxycholate, 150 mM NaCl, 1 mM EGTA, 1 mmol/l Na3VO4, 1 mM NaF, and protease inhibitors (1 μg/mL aprotinin, 10 μg/mL leupeptin, and 1 mM 4-(2-aminoethyl) benzenesulfonyl fluoride)) for 2 h on ice, as described earlier [32,39]. Protein extracts were cleared by centrifugation, and the protein concentrates were properly harvested. Thereafter, the protein concentration was determined using the Lowry protein assay [40], and immediately, this sample was combined at equal volume with Laemmli buffer (2X sodium dodecyl sulfate (SDS), 100 mM Tris-HCl (pH 6.8), 200 mM dithiothreitol (DTT), 4% SDS, 0.1% bromophenol blue, and 20% glycerol). The proteins were resolved into SDS–PAGE (8% or 10%) and afterward transferred to PVDF membranes (Millipore, MA, USA), which were properly blocked with 1% bovine serum albumin (2.5%) in Tris-buffered saline (TBS)–Tween-20 (0.05%), and incubated overnight with an appropriate primary antibody at 1:1000 dilutions. After washing in TBS–Tween-20 (0.05%), membranes were incubated with secondary conjugated anti-rabbit, anti-goat, or anti-mouse IgGs antibodies at 1:5000 dilutions (all in immunoblotting assays) in blocking buffer for 1 h. Immunoreactive bands were detected with an enhanced chemiluminescence (ECL) kit (Thermo Scientific, MA, USA).

RNA extraction and qPCR analysis: For total mRNA extraction, HUVEC cells previously treated with conditioned medium for up to 72 h were harvested with PBS and immediately homogenized with 0.5 mL of Ambion TRIzol Reagent (Life Sciences-Fisher Scientific Inc., Waltham, MA, USA). The total RNA was extracted using the TRIzol/chloroform protocol. After RNA extraction, the concentration and purity were determined using a microplate reader (SYNERGY-HTX multi-mode reader, Biotek, Tigan St, Winooski, VT, USA). For the gene expression study, first, cDNA synthesis was performed with a high-capacity cDNA reverse transcription kit (Applied Biosystems, Foster City, CA, USA) according to the manufacturer’s instructions after DNase I treatment (Invitrogen, Carlsbad, CA, USA). Posteriorly, qPCR reactions were carried in a QuantStudio^®^3 Real-Time PCR (Thermo Fisher Scientific, Waltham, MA, USA) in reactions of 10 μL Syber Green Master Mix 2x-5 μL, 0.4 μM of each primer (for primers and conditions, see Table 1), 50 ng of cDNA, and nuclease-free H_2_O. The GAPDH gene was considered a housekeeping gene.

Statistical analysis: Densitometry analysis of the immunoblots bands was performed, and the arbitrary values were represented as mean ± standard deviation (SD). They were verified using Student’s *t*-test (2-tailed) with *p* < 0.05 considered statistically significant and *p* < 0.001 considered highly significant. In an experiment where there were >2 groups, we used one-way ANOVA (non-parametric) with a Bonferroni post-test in order to compare all pairs of groups. In this case, the significance level was considered to be reached when α = 0.05 (95% confidence interval). The software used was GraphPad Prism version 6.0.

## 3. Results

### 3.1. Experimental Design

To assess the effect of Co-Cr on endothelial cells (HUVEC), an indirect experimental model was explored where cells received a medium previously conditioned by Co-Cr for up to 72 h (Figure 1a–d). First, we investigated the expression of genes and proteins related to enzymes that catalyze the addition (DNMTs) or removal (TETs) of a methyl group in a DNA strand in a process called DNA methylation metabolism (Figure 1e). Furthermore, the mRNA transcription mechanism can be regulated by another epigenetic marker responsible for promoting the acetylation (HATs) or deacetylation (HDACs) of histones (Figure 1g) in lysine residues in the N-terminal branch. Together, these mechanisms make the DNA strand accessible to RNA polymerase, which promotes gene transcription (Figure 1h).

### 3.2. Effect of Co-Cr-Enriched Medium on JNK Phosphorylation

Figure 2 reveals the involvement of c-Jun N-terminal kinase (JNK) activation in response to the Co-Cr-enriched medium. Importantly, it seems that JNK Phosphorylation (Thr180/Tyr182) is constitutively required in endothelial cells once it becomes phosphorylated with the culture groups investigated in this study, although a significant decrease has been found in response to Co-Cr-enriched medium.

### 3.3. Effect of Co-Cr-Enriched Medium on the Protein Content of Histone Deacetylase Enzymes

Additionally, for the analysis of the involvement of the methylation metabolism in DNA strands, we focused on understanding whether there is involvement with histone acetylation metabolism in endothelial cells responding to Co-Cr and if it is relevant and complimentary to previous findings looking to map the biological effect during angiogenesis. As previously reported here, histone deacetylases (HDACs) are responsible for promoting the lysine breakdown of the N-terminal region of histones and driving gene transcription. Our data show a higher expression (mRNA) of HDAC1 (Figure 3a), which also reflects the protein content (Figure 3(a’,a”)) in response to both Co-Cr_wo and Co-Cr_w, while HDAC2-related expression (Figure 3b) and protein content (Figure 3(b’,b”)) was downregulated in response to the Co-Cr medium.

Additionally, both HDAC3 (Figure 4a) and HDAC6 (Figure 4b) behaviors were also investigated in this study in endothelial cells responding to Co-Cr_wo and Co-Cr_w. Our data show both HDACs genes were higher in the Co-Cr-enriched medium (Figure 4a,b), but it was reflected only in the increase in the HDAC6 protein profile (Figure 4b(’,b”)), while the HDAC3 protein content was lower in endothelial cells responding to the Co-Cr medium (Figure 4(a’,a”)).

Furthermore, Sirtuin 1 (SIRT1), another protein member of the histone deacetylase family, was also addressed in this study. Our data clearly show that both Co-Cr conditions (Co-Cr_wo and Co-Cr_w) positively modulate the expression of SIRT1 in exposed endothelial cells (Figure 5), fully considering mRNA (Figure 5a) and protein content (Figure 5b,c).

### 3.4. DNA Methylation-Related DNMTs Were Investigated in Response to Co-Cr Alloys

The DNA methylation enzyme family mainly consists of DNMT1, DNMT3A, and DNMT3B, and all of them were analyzed in this study. Specifically, we observed that DNMT1 is differentially expressed in response to each condition of the Co-Cr alloy investigated in this study. Regarding DNMT1, the HUVEC cells exposed to both Co-Cr conditions responded differentially; considering the qPCR technology, the expression of this gene was lower than the control in HUVECs responding to Co-Cr_wo and in an opposite way responding to Co-Cr_w (Figure 6a), while in the protein content for DNMT1, the cells treated with Co-Cr_w seemed to decrease its profile (Figure 6(a’,a”)).

Regarding DNMT3A, the mRNA showed that there were no differences in cells responding to the Co-Cr_w medium, while in the cells that received the Co-Cr_wo-conditioned medium, the profile was higher (Figure 6b); considering the protein profile of DNMT3A, it showed a very similar response with DNMT1, where the group that received the Co-Cr_w medium showed a decrease in its protein content (Figure 6(b’,b”)). Furthermore, we evaluated the effects of Co-Cr-enriched media on DNMT3B expression (mRNA); we found that there were no significances between groups treated with different media in relation to the control (Figure 6c). However, the protein levels showed a significative increase in the Co-Cr_wo and Co-Cr_w groups when compared with the control (Figure 6(c’,c”)).

### 3.5. DNA Demethylation-Related TETs Were Investigated in Response to Co-Cr Alloys

TETs are enzymes responsible for removing the methyl group from DNA strands, which were investigated here (Figure 7). Comparing the behavior of TET1, we verified that the Co-Cr_w-conditioned medium promoted the higher expression of mRNA (~four-fold changes; Figure 7a), but it does not reflect the protein content in response to both groups of cells exposed to Co-Cr (Figure 7(a’,a”)). Performing a very similar profile, TET2 (mRNA) was higher in response to the Co-Cr_w group (~15-fold changes; Figure 7b) compared with the control, while the protein content was higher only in response to the Co-Cr_wo group (Figure 7(b’,b”)). Altogether, it seems clear that Co-Cr requires a repertoire of genes and proteins related to epigenetic machinery in endothelial cells, and it opens new perspectives to better know their effects on the functional performance of blood vessels during angiogenesis responding to Co-Cr during its osseointegration.

## 4. Discussion

Epigenetic mechanisms control the levels of gene expression and the possible phenotype changes in cells [32,41]. Specifically, epigenetic metabolism is performed by the mechanisms of acetylation, methylation, micro-RNA (miRNAs), and long non-coding RNA (lnc-RNAs) controls [42,43,44,45,46]. As angiogenesis is widely known to be decisive during bone oppositional growth during the osseointegration of biomaterials in dentistry and medicine, the goal here was to better address the epigenetic mechanism, which is able to drive the phenotype of endothelial cells responding to Co-Cr. Although there are robust reports in the literature about its physicochemical properties regarding Co-Cr, its effect on epigenetic mechanisms in endothelial cells is barely known.

Our data show that Co-Cr requires DNMTs, specifically, DNMT1, which is known to be an important player during the S-phase of the cell cycle [47,48,49], while DNMT3A and DNMT3B are related during new DNA strand methylation [50,51]. Among DNMTs, Co-Cr requires significant DNMT3B involvement, as well as both TET1 and TET2, which may result in an increase in DNA demethylation. Conversely, the balance between DNMTs and TETs determines the methylation profile of DNA strands in eukaryotic cells. It is important to mention that although there is an expected proportional translation of mRNA transcripts into protein, our data bring special attention to this, which opens up the possibility of action from posttranscriptional-based mechanisms requiring miRNAs and lnc-RNAs. Another point to be considered here is the ability of proteasomes to turn over intracellular proteins such as those DNMTs and TETs. Altogether, this mechanism is new to the literature, mainly considering the potential effect of cobalt in promoting a hypoxia condition [52], and it might contribute to endothelial cell proliferation and, thus, coordinate angiogenesis. Therefore, the repercussion of this condition needs to be better investigated by looking at the behavior of epigenetic players related to cytoskeleton rearrangement, such as HDACs, which are able to modulate the polymerization of microtubules and were also discussed during cell cycle progression [53,54,55]. In this context, HDAC6 seems to have a more significant effect on endothelial cells responding to Co-Cr, and it might be related to survival and proliferative molecular mechanisms related to endothelial cells [56]. Among them, JNK is related to important cellular survival signaling that involves cytoskeleton rearrangement [57].

Finally, the requirement of SIRT1 was also another very interesting datum obtained in this study. SIRT1 is associated with a capacity to modulate the expression of HIF1A in response to hypoxia microenvironments, thus presenting a protective effect. Additionally, SIRT1, a mammalian homolog of yeast silent information regulator 2 (Sir2), is a survival factor that is involved in lifespan extension, and recently, it has been reported with respect to HIF1A, as it deacetylates its lysine residues. It is important to declare that, as SIRT1 is a druggable molecule, it might be assembled in biotechnological platforms to predict the response of cells to biomaterials. Of note, HIF1A, a transcription factor that mediates the crosstalk between angiogenesis and osteogenesis, is expected during bone growth.

## 5. Conclusions

Altogether, our study brings a descriptive repertoire of molecules related to epigenetic metabolism in endothelial cells responding to Co-Cr, and it opens new perspectives to investigate whether this mechanism might affect blood vessel-related properties considering cell adhesion, cell cycle progression, and the sprouting of the vessels during the osteointegration mechanism of implantable devices. Furthermore, it brings some light to understanding the failure of bone implants in specific pathophysiological conditions where a lack of angiogenesis is found.

## Figures and Tables

**Figure 1 jfb-14-00127-f001:**
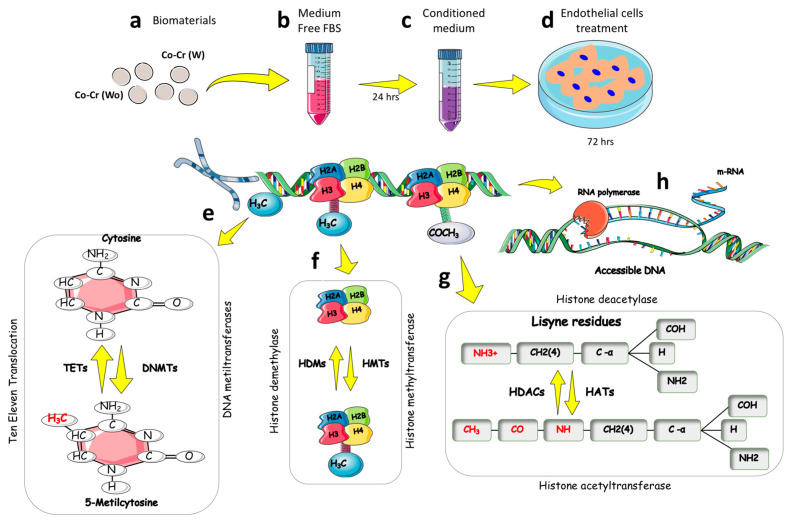
Experimental design. Co-Cr alloys without etching (wo) (**a**) and with DAE (w) conditioned the culture medium for 24 h (**b**). Medium was collected (**c**) to treat endothelial cells for up to 72 h (**d**). Endothelial cells were properly lysed and we proceeded with real-time qPCR technology or immunoblotting analyses. The enzymes responsible for the addition (DNMTs) or removal of the methyl group (TETs) in the CpG islands of the DNA (**e**), as well as enzymes responsible for the removal (HDACs) of lysine in the N-terminal tail of the histones (**g**) were investigated here. The scheme depicts an important epigenetic regulation mechanism (**f**) responsible for promoting histone methylation. In (**h**), it is possible to observe the trigger of the molecular regulatory process by involving epigenetic marks where the DNA is accessible for use via the transcription of the messenger RNA.

**Figure 2 jfb-14-00127-f002:**
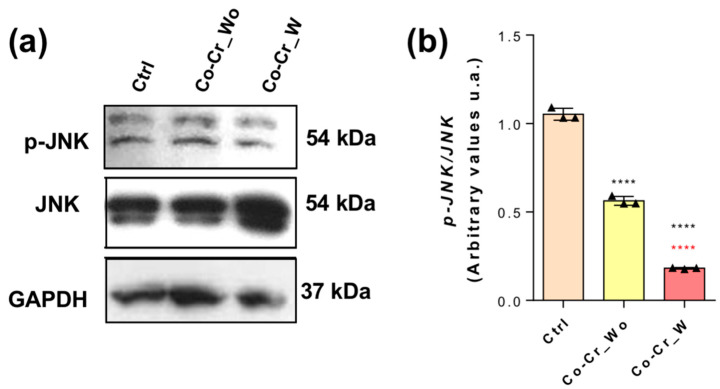
MAPK-JNK seems to play a constitutive role in endothelial cells. After 72h of exposition with the different Co-Cr conditions, the endothelial cells were lysed in a specific buffer to study the protein content using a Western blot assay. Proteins (75 µg) were resolved into SDS-PAGE (10%). Both JNK (**a**) and p-JNK (Thr180/Tyr182) (**b**) were investigated. As housekeeping control, GAPDH was used to normalize the data. The graphs represent the experimental analysis, *n* = 3. **** *p* < 0.0001: Statistical difference when compared with the control group. **** *p* < 0.0001: Statistical difference when compared with Co-Cr_wo and Co-Cr_w.

**Figure 3 jfb-14-00127-f003:**
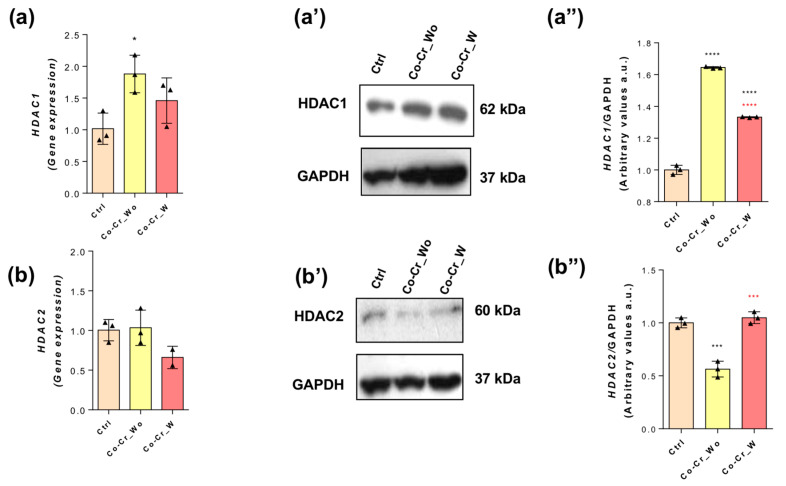
Effect of Co-Cr-enriched medium on HDAC1 and HDAC2. To investigate the modulation profile of HDACs 1 and 2, endothelial cells were properly lysed to allow for the analysis of the mRNA via qPCR: HDAC1 (**a**) and HDAC2 (**b**). Protein content found via Western blot (WB) technology: HDAC1 (**a’,a”**) and HDAC2 (**b’,b”**). * *p* < 0.05; *** *p* < 0.0002; **** *p* < 0.0001: Statistical difference when compared with the control group. *** *p* < 0.0002; **** *p* < 0.0001: Statistical differences when compared with Co-Cr_wo and Co-Cr_w. GAPDH was used as an internal control for sample loading (WB) and as a housekeeping gene (qPCR). The experiments were performed in triplicate (*n* = 3).

**Figure 4 jfb-14-00127-f004:**
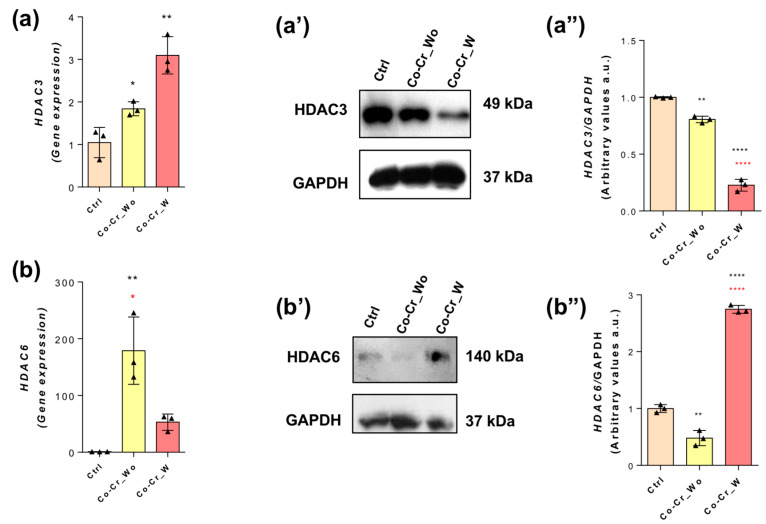
HDAC6 seems to play a role in Co-Cr-enriched medium. To investigate the modulation profile of HDACs 3 and 6, endothelial cells were properly lysed to allow for the analysis of the mRNA by qPCR: HDAC1 (**a**) and HDAC2 (**b**). Protein content found via Western blot (WB) technology: HDAC1 (**a’,a”**) and HDAC2 (**b’,b”**). * *p* < 0.05; ** *p* < 0.0006; **** *p* < 0.0001: Statistical differences when compared with the control group. * *p* < 0.05; **** *p* < 0.0001: Statistical differences when compared with Co-Cr_wo and Co-Cr_w. GAPDH was used as an internal control for the sample loading (WB) and as a housekeeping gene (qPCR). The experiments were performed in triplicate (*n* = 3).

**Figure 5 jfb-14-00127-f005:**
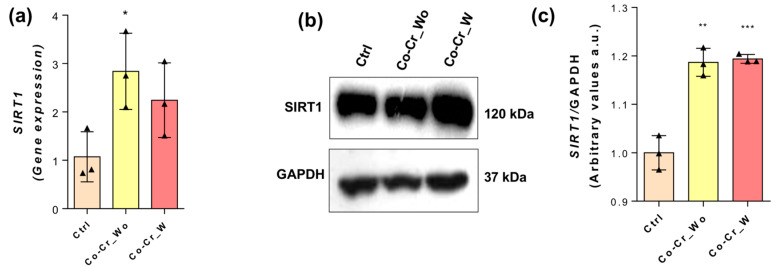
Epigenetic mechanism requires SIRT1 in response to Co-Cr. To investigate the modulation profile of SIRT1, endothelial cells were properly lysed to allow for the analysis of the mRNA via qPCR (**a**), and protein content was determined using Western blot (**b**,**c**). Statistics: * *p* < 0.05; ** *p* < 0.0006; *** *p* < 0.0002: Statistical differences when compared with the control group. GAPDH was used as an internal control for sample loading (WB) and as a housekeeping gene (qPCR). The experiments were performed in triplicate (*n* = 3).

**Figure 6 jfb-14-00127-f006:**
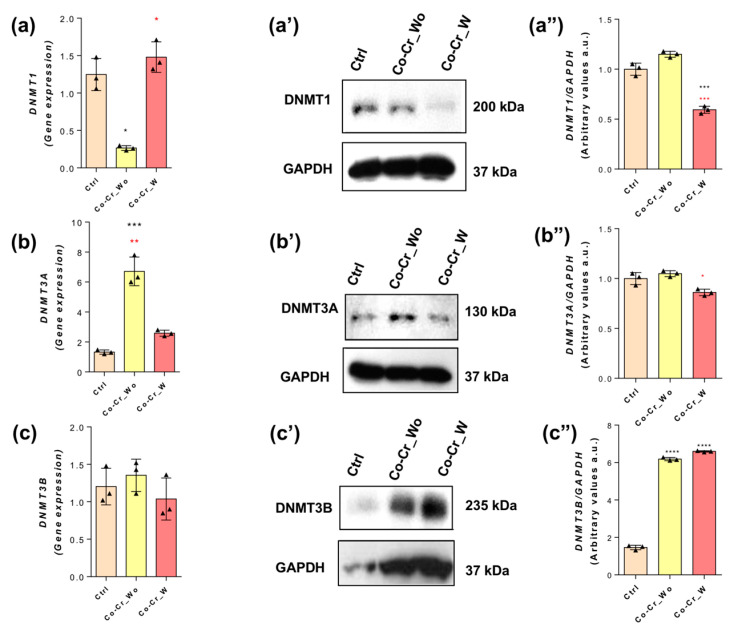
DNMT3B is required for endothelial cells to respond to Co-Cr. The expression levels and protein content of DNMT1 (**a**), DNMT3A (**b**), and DNMT3B (**c**) were investigated in the endothelial cells responding to Co-Cr_wo and Co-Cr_w, while the protein content for DNMT1 is depicted in (**a’**, representative image) and (**a”**, densitometry analysis), and DNMT3A is depicted in (**b’**, representative image) and (**b”**, densitometry analysis), and DNMT3B is depicted in (**c’**, representative image) and (**c”**, densitometry analysis). Statistics: * *p* < 0.05; *** *p* < 0.0002; **** *p* < 0.0001: Statistical differences when compared with the control group. * *p* < 0.05; ** *p* < 0.0004; *** *p* < 0.0003: Statistical differences when compared with Co-Cr_wo and Co-Cr_w. GAPDH was used as an internal control for sample loading (WB) and as a housekeeping gene (qPCR). The experiments were performed in triplicate (*n* = 3).

**Figure 7 jfb-14-00127-f007:**
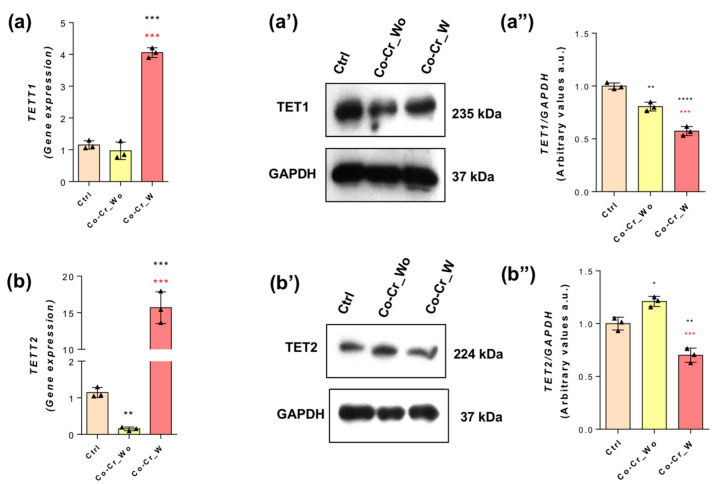
TET involvement in endothelial cells responding to Co-Cr alloys. The conditioned medium was conditioned using two different Co-Cr alloy surfaces: Co-Cr_wo (without DAE) and Co-Cr_w (with DAE), thereafter used to expose endothelial cells after up to 72 h, when the samples were properly harvested to allow for an mRNA analysis using qPCR technology and to determine protein content using Western blot (WB). The profiles of TET1 and TET2mRNA are displayed in (**a**) and (**b**), while the protein content for TET1 is depicted in (**a’**, representative image) and (**a”**, densitometry analysis), and TET2 is depicted in (**b’**, representative image) and (**b”**, densitometry analysis). GAPDH was used as an internal control for sample loading (WB) and as a housekeeping gene (qPCR). The experiments were performed in triplicate (*n* = 3). Statistics: * *p* < 0.05; ** *p* < 0.0006; *** *p* < 0.0002; **** *p* < 0.0001: Statistical differences when compared with the control group. *** *p* < 0.0002: Statistical differences when compared with Co-Cr_wo and Co-Cr_w. Note: TET methylcytosine dioxygenase (TET).

**Table 1 jfb-14-00127-t001:** Primer sequences and PCR cycle conditions.

Gene (ID)	Primer	5′–3′ Sequence	Reactions Condition
HDAC1 (3065)	Forward	CTGGCCATCATCTCCTTGAT	95 °C-15 s; 58 °C-30 s; 72 °C-60 s
	Reverse	ACCAGAGACGTGGAAACTGG	
HDAC2 (3066)	Forward	TTCTCAGTGCACCCAGTCAG	95 °C-15 s; 58 °C-30 s; 72 °C-60 s
	Reverse	CCAGTATCCTTGGGGGAAAT	
HDAC3 (8841)	Forward	ACGTGGGCAACTTCCACTAC	95 °C-15 s; 58 °C-30 s; 72 °C-60 s
	Reverse	GACTCTTGGTGAAGCCTTGC	
HDAC6 (10013)	Forward	AAGTAGGCAGAACCCCCAGT	95 °C-15 s; 58 °C-30 s; 72 °C-60 s
	Reverse	GTGCTTCAGCCTCAAGGTTC	
SIRT1 (23411)	Forward	GCAGATTAGTAGGCGGCTTG	95 °C-15 s; 58 °C-30 s; 72 °C-60 s
	Reverse	TCTGGCATGTCCCACTATCA	
DNMT1 (1786)	Forward	AGGACCCAGACAGAGAAGCA	95 °C-15 s; 58 °C-30 s; 72 °C-60 s
	Reverse	GTACGGGAATGCTGAGTGGT	
DNMT3A (13435)	Forward	AGGAAGCCCATCCGGGTGCTA	95 °C-15 s; 58 °C-30 s; 72 °C-60 s
	Reverse	AGCGGTCCACTTGGATGCCC	
DNMT3B (1789)	Forward	TCGACTTGGTGGTTATTGTCTG	95 °C-15 s; 58 °C-30 s; 72 °C-60 s
	Reverse	TCGAGCTACAAGACTGCTTGG	
TET1 (80312)	Forward	GCCCCTCTTCATTACCAAGTC	95 °C-15 s; 58 °C-30 s; 72 °C-60 s
	Reverse	CGCCAGTTGCTTATCAAAATC	
TET2 (54790)	Forward	GGTGCCTCTGGAGTGACTGT	95 °C-15 s; 58 °C-30 s; 72 °C-60 s
	Reverse	GGAAAATGCAAGCCCTATGA	
TET3 (200424)	Forward	GGTCAGGCTGGTTTACAACG	95 °C-15 s; 58 °C-30 s; 72 °C-60 s
	Reverse	GGCATAGACCCACACACATCT	
GAPDH (2597)	Forward	AAGGTGAAGGTCGGAGTCAA	95 °C-15 s; 58 °C-30 s; 72 °C-60 s
	Reverse	AATGAAGGGGTCATTGATGG	

## Data Availability

The data that support the findings of this study are available from the corresponding author upon reasonable request.

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
