# Peer review of "Epigenetic Differences Arise in Endothelial Cells Responding to Cobalt–Chromium"

_jfb, 2023, doi:10.3390/jfb14030127_

Round 1

Reviewer 1 Report

Comments on Fernandes et al:

The aim of this manuscript is to investigate how epigenetic metabolism drive the proliferative phenotype of endothelial cells towards Co-Cr, in order to better define its role in cell adhesion, cell cycle and angiogenesis, surrounding implantable devices.

This manuscript shows rich content, providing a deep insight for some works: the study is within the journal’s scope, and I found it to be well-written, providing sufficient information. Even if the manuscript provides an organic overview, with a densely organized structure and based on well-synthetized evidence, there are some suggestions necessary to make the article complete and fully readable. For these reasons, the manuscript requires major changes.

Please find below an enumerated list of comments on my review of the manuscript:

INTRODUCTION:

LINE 37: Biomaterials are a class of materials, characterized by chemical, mechanical, and biological properties, which make them suitable for contact with living tissues (see, for reference: Bianchi, S., Bernardi, S., Mattei, A., Cristiano, L., Mancini, L., Torge, D., ... & Marchetti, E. (2022). Morphological and Biological Evaluations of Human Periodontal Ligament Fibroblasts in Contact with Different Bovine Bone Grafts Treated with Low-Temperature Deproteinisation Protocol. International Journal of Molecular Sciences23(9), 5273). In this introductive section, authors should provide a complete and organic definition of biomaterial, also mentioning the main features of these materials and highlighting their application in regenerative procedures.

LINE 61: Before describing how epigenetics drive endothelial cells’ phenotype, the authors should provide a complete definition of the main epigenetic modification. For example, DNA methylation is the addiction of a methyl group to a DNA nucleotide, with a prominent role in transcriptional regulation, specifically as regard the silencing of a specific gene. Moreover, demethylation, which consists of the removal of a methyl group from a DNA nucleotide, may control imprinted gene expression; at the same time, the most important carrier of epigenetic information is the post- translational modification of histones, which have emerged as one of the prime constituents of the transcription regulatory machinery. Among them, the methylation and acetylation of lysine residues, although lysine ubiquitylation and serine/threonine/tyrosine phosphorylation, as main means of epigenetic control (see, for reference: Fitz-James, M. H., & Cavalli, G. (2022). Molecular mechanisms of transgenerational epigenetic inheritance. Nature Reviews Genetics23(6), 325-341). The authors should provide an organic and brief description of the main epigenetics pathways, in order to include complete information for expert and non – expert in the field.

The main topic is interesting, and certainly of great clinical impact. As regards the originality and strengths of this manuscript, this is a significant contribute to the ongoing research on this topic, as it extends the research field on the regulatory role of epigenetic metabolism in cell adhesion, cell cycle and angiogenesis, surrounding implantable devices. Overall, the contents are rich, and the authors also give their deep insight for some works.

As regards the section of methods, there is a specific and detailed explanation for the methods used in this study: this is particularly significant, since the manuscript relies on a multitude of methodological and statistical analysis, to derive its conclusions. The methodology applied is overall correct, the results are reliable and adequately discussed.

The conclusion of this manuscript is perfectly in line with the main purpose of the paper: the authors have designed and conducted the study properly. As regards the conclusions, they are well written and present an adequate balance between the description of previous findings and the results presented by the authors.

Finally, this manuscript also shows a basic structure, properly divided and looks like very informative on this topic. Furthermore, figures and tables are complete, organized in an organic manner and easy to read.

In conclusion, this manuscript is densely presented and well organized, based on well-synthetized evidence. The authors were lucid in their style of writing, making it easy to read and understand the message, portrayed in the manuscript. Besides, the methodology design was appropriately implemented within the study. However, many of the topics are very concisely covered. This manuscript provided a comprehensive analysis of current knowledge in this field. Moreover, this research has futuristic importance and could be potential for future research. However, major concerns of this manuscript are with the introductive section: for these reasons, I have major comments for this section, for improvement before acceptance for publication. The article is accurate and provides relevant information on the topic and I have some major points to make, that may help to improve the quality of the current manuscript and maximize its scientific impact. I would accept this manuscript if the comments are addressed properly.

Author Response

Feb 15, 2023.

Dear Editor:

Thank you for the attention you have given to our manuscript originally called “Cobalt-Chromium alloys requires epigenetic machinery to drive the proliferative phenotype of endothelial cells” (jfb-2197692). Indeed, we appreciated the concerns and criticism of the reviewers and the opportunity you have given us to submit a revision of our reworked manuscript.

Importantly, we addressed all the points raised in full, and feel that by incorporating their suggestions our manuscript has been much improved.

Based on the significant changes done in this study, a new title is now suggested as follows: “Epigenetic differences arise in endothelial cell responding to cobalt-chromium”

Our responses to the raised issues are summarized as follows:

REVIEWER 1:

The aim of this manuscript is to investigate how epigenetic metabolism drive the proliferative phenotype of endothelial cells towards Co-Cr, in order to better define its role in cell adhesion, cell cycle and angiogenesis, surrounding implantable devices.

This manuscript shows rich content, providing a deep insight for some works: the study is within the journal’s scope, and I found it to be well-written, providing sufficient information. Even if the manuscript provides an organic overview, with a densely organized structure and based on well-synthetized evidence, there are some suggestions necessary to make the article complete and fully readable. For these reasons, the manuscript requires major changes.

Please find below an enumerated list of comments on my review of the manuscript:

INTRODUCTION:

LINE 37: Biomaterials are a class of materials, characterized by chemical, mechanical, and biological properties, which make them suitable for contact with living tissues (see, for reference: Bianchi, S., Bernardi, S., Mattei, A., Cristiano, L., Mancini, L., Torge, D., ... & Marchetti, E. (2022). Morphological and Biological Evaluations of Human Periodontal Ligament Fibroblasts in Contact with Different Bovine Bone Grafts Treated with Low-Temperature Deproteinisation Protocol. International Journal of Molecular Sciences, 23(9), 5273). In this introductive section, authors should provide a complete and organic definition of biomaterial, also mentioning the main features of these materials and highlighting their application in regenerative procedures.

Reaction: Dear reviewer, thanks for your time and attention in carefully analyzing our study. We have now implemented all of these concerns properly. This new version of this study brings better discussion on the main findings by the authors. Furthermore, a carefully effort was dedicated on the introduction by serving from your criticism and knowledge.

LINE 61: Before describing how epigenetics drive endothelial cells’ phenotype, the authors should provide a complete definition of the main epigenetic modification. For example, DNA methylation is the addiction of a methyl group to a DNA nucleotide, with a prominent role in transcriptional regulation, specifically as regard the silencing of a specific gene. Moreover, demethylation, which consists of the removal of a methyl group from a DNA nucleotide, may control imprinted gene expression; at the same time, the most important carrier of epigenetic information is the post- translational modification of histones, which have emerged as one of the prime constituents of the transcription regulatory machinery. Among them, the methylation and acetylation of lysine residues, although lysine ubiquitylation and serine/threonine/tyrosine phosphorylation, as main means of epigenetic control (see, for reference: Fitz-James, M. H., & Cavalli, G. (2022). Molecular mechanisms of transgenerational epigenetic inheritance. Nature Reviews Genetics, 23(6), 325-341). The authors should provide an organic and brief description of the main epigenetics pathways, in order to include complete information for expert and non – expert in the field.

Reaction. As requested, it was better articulated along the introduction section.

The main topic is interesting, and certainly of great clinical impact. As regards the originality and strengths of this manuscript, this is a significant contribute to the ongoing research on this topic, as it extends the research field on the regulatory role of epigenetic metabolism in cell adhesion, cell cycle and angiogenesis, surrounding implantable devices. Overall, the contents are rich, and the authors also give their deep insight for some works.

As regards the section of methods, there is a specific and detailed explanation for the methods used in this study: this is particularly significant, since the manuscript relies on a multitude of methodological and statistical analysis, to derive its conclusions. The methodology applied is overall correct, the results are reliable and adequately discussed.

The conclusion of this manuscript is perfectly in line with the main purpose of the paper: the authors have designed and conducted the study properly. As regards the conclusions, they are well written and present an adequate balance between the description of previous findings and the results presented by the authors.

Finally, this manuscript also shows a basic structure, properly divided and looks like very informative on this topic. Furthermore, figures and tables are complete, organized in an organic manner and easy to read.

In conclusion, this manuscript is densely presented and well organized, based on well-synthetized evidence. The authors were lucid in their style of writing, making it easy to read and understand the message, portrayed in the manuscript. Besides, the methodology design was appropriately implemented within the study. However, many of the topics are very concisely covered. This manuscript provided a comprehensive analysis of current knowledge in this field. Moreover, this research has futuristic importance and could be potential for future research. However, major concerns of this manuscript are with the introductive section: for these reasons, I have major comments for this section, for improvement before acceptance for publication. The article is accurate and provides relevant information on the topic and I have some major points to make, that may help to improve the quality of the current manuscript and maximize its scientific impact. I would accept this manuscript if the comments are addressed properly.

Reaction. Particularly, I must thank you for all the teachings along the preparation of the reworked manuscript.

In conclusion, we feel that we have taken into account the concerns of the reviewer in full and that the present version of the manuscript should be acceptable for publication in Journal of Functional Materials.

Looking forward to your evaluation,

WILLIAN FERNANDO ZAMBUZZI

On behalf the authors

Reviewer 2 Report

Reviewer Comments:

In this manuscript, the authors make an interesting observation about the involvement of epigenetic metabolism necessary to derive the proliferative phenotype of endothelial cells in response to Co-Cr. Here, authors demonstrated that methylation profile in response to Co-Cr modulated by DNMTs (DNA Methyltransferase) mainly DNMT3B and both TETs i.e., TET1 and TET2. Once Co-Cr is associated with the hypoxia condition, the cell requires MAPKp38.

In its current format, the manuscript is preliminary, the data does not fully support the conclusions and the results are premature and overstated. The manuscript needs major revision and language editing before acceptance. Several concerns are highlighted below.

Comment 1: Fig 2. In Figures 2A, 2B, and 2C, the level of native ERK1/2 and JNK1/2 increases with Co—Cr treatment whereas p38 decreases. The level of p-p38 level and native p38 level also decreases with Co—Cr treatment. Ideally, for the activation or inhibition of the MAPK pathway, the p-p38 level should increase or decrease without a change in the level of native p38 protein. The overall data doesn’t mean anything, and the quality of the blot is not good. Quantification data doesn’t align with p-p38 level alteration in western blot.

Comment 2: Fig 3. In Fig 3g, 3h, and 3I, there is no significant increase in DNMT3B gene expression levels whereas there is a 3-fold increase in DNMT3B at the protein level. Ideally, an increase in the level of DNMT3B gene expression should align with a fold increase in DNMT3B at the protein level after Co-Cr treatment. Please replace with a better representative blot image for DNMT1, DNMT3A, and DNMT3B with better loading control. The overall data look confusing and misguiding. Difficult to conclude anything.

Comment 3: Fig 4. Alteration of TET1 and TET2 pattern at RNA level in response to Co-Cr treatment doesn’t align with pattern of change of TET1 and TET2 at the protein level. This cannot be explained by as stated by the author i.e.  negative feedback between transcripts 226 (mRNA) and proteins. It looks like the experiment is not done in the control condition. Data need to be improved.    

Comment 4: Fig 5. The quality of the blot is poor. The quantification graph doesn’t align with blot intensity and pattern of change. HDAC6 gene expression is completely different as compared to HDAC6 at the protein level. I am wondering how an author will explain this.

Comment 5: Fig 6. SIRT1 and HMGB1 blot is saturated and the gene expression pattern for both SIRT1 AND HMGB1 doesn’t align with western blot data and quantification.

Author Response

Feb 15, 2023.

Dear Editor:

Thank you for the attention you have given to our manuscript originally called “Cobalt-Chromium alloys requires epigenetic machinery to drive the proliferative phenotype of endothelial cells” (jfb-2197692). Indeed, we appreciated the concerns and criticism of the reviewers and the opportunity you have given us to submit a revision of our reworked manuscript.

Importantly, we addressed all the points raised in full, and feel that by incorporating their suggestions our manuscript has been much improved.

Our responses to the raised issues are summarized as follows:

REVIEWER 2:

In this manuscript, the authors make an interesting observation about the involvement of epigenetic metabolism necessary to derive the proliferative phenotype of endothelial cells in response to Co-Cr. Here, authors demonstrated that methylation profile in response to Co-Cr modulated by DNMTs (DNA Methyltransferase) mainly DNMT3B and both TETs i.e., TET1 and TET2. Once Co-Cr is associated with the hypoxia condition, the cell requires MAPKp38. In its current format, the manuscript is preliminary, the data does not fully support the conclusions and the results are premature and overstated. The manuscript needs major revision and language editing before acceptance. Several concerns are highlighted below.

Reaction: Dear reviewer, thanks for the attention in analyzing our study. We have now implemented all of these concerns properly. This new version of this study brings better discussion on the main findings by the authors. Furthermore, a carefully language editing as requested and because that please accept our apologies by the inconvenience.

With the significant changes done in this study, a new title is now suggested as follows: “Epigenetic differences arise in endothelial cell responding to cobalt-chromium”

Comment 1: Fig 2. In Figures 2A, 2B, and 2C, the level of native ERK1/2 and JNK1/2 increases with Co—Cr treatment whereas p38 decreases. The level of p-p38 level and native p38 level also decreases with Co—Cr treatment. Ideally, for the activation or inhibition of the MAPK pathway, the p-p38 level should increase or decrease without a change in the level of native p38 protein. The overall data doesn’t mean anything, and the quality of the blot is not good. Quantification data doesn’t align with p-p38 level alteration in western blot.

Reaction. Many thanks to bring this up. We were convinced from this analysis in removing out the data about p38 and Erk from the manuscript.

Comment 2: Fig 3. In Fig 3g, 3h, and 3I, there is no significant increase in DNMT3B gene expression levels whereas there is a 3-fold increase in DNMT3B at the protein level. Ideally, an increase in the level of DNMT3B gene expression should align with a fold increase in DNMT3B at the protein level after Co-Cr treatment. Please replace with a better representative blot image for DNMT1, DNMT3A, and DNMT3B with better loading control. The overall data look confusing and misguiding. Difficult to conclude anything.

Reaction. Once again, many thanks for bringing this up. Unfortunately, we have performed the western blot using classical x-ray developer and the images aren’t with good quality as those usually acquired using a gel/membrane documentation system. That’s a pity! However, we believe the ratio of the arbitrary values obtained from the densitometry with that ones rigorously normalized using the loading control (GADPH) brings reliable data to understanding the effect of the Co-Cr on endothelial cells.

Regarding the transcripts profile of target genes does not align with protein level is quite explainable considering the central molecular biology dogma where the posttranscriptional mechanisms governing the mature mRNA synthesis is compromised with the roles of microRNAs as well as lncRNAs in this way. From this perspective, we intend to better address this issue in near future by looking for understanding the potential posttranscriptional mechanism related with epigenetic matters. This is now better discussed in the reworked manuscript. Hope you understand this manuscript brings a very preliminary descriptive data about epigenetic machinery involved with the biological response to Co-Cr and something about mechanisms will be addressed further.

Comment 3: Fig 4. Alteration of TET1 and TET2 pattern at RNA level in response to Co-Cr treatment doesn’t align with pattern of change of TET1 and TET2 at the protein level. This cannot be explained by as stated by the author i.e.  negative feedback between transcripts 226 (mRNA) and proteins. It looks like the experiment is not done in the control condition. Data need to be improved.

Reaction: Dear reviewer, thanks for the attention in analyzing our study with this criticism, make sure it promoted a huge update in our manuscript. Once again, a probable explanation on this regard covers all it was said in the comment 2. Hope this is fine for you.

Comment 4: Fig 5. The quality of the blot is poor. The quantification graph doesn’t align with blot intensity and pattern of change. HDAC6 gene expression is completely different as compared to HDAC6 at the protein level. I am wondering how an author will explain this.

Reaction: Dear reviewer, sorry for this inconvenience. The quality of blots is not fit with the quality control when the images are obtained from gel/membrane documentation system; however, we believe this is enough to respond whether epigenetics players are involved with CoCr responses.

Comment 5: Fig 6. SIRT1 and HMGB1 blot is saturated and the gene expression pattern for both SIRT1 AND HMGB1 doesn’t align with western blot data and quantification.

Reaction. The quantification of the bands is normalized with the densitometry of those bands related with the loading control (in this case, GADPH). The proportional alignment considering transcripts and proteins needs observe other epigenetics’-based modulations as it was explained earlier in this letter. Make sure that all of these concerns were taken into consideration to rework the manuscript – since blots analysis/organization to better present the subjects and subtitles along the text. In this version, we decided to show the data using a dotplot configuration of the graph. On behalf the authorship, thank you! Your comments became better our study’s documentation. Hope this is fine for you at this reworked version of the manuscript.

In conclusion, we feel that we have taken into account the concerns of the reviewer in full and that the present version of the manuscript should be acceptable for publication in Journal of Functional Materials.

Looking forward to your evaluation,

WILLIAN FERNANDO ZAMBUZZI

On behalf the authors

Reviewer 3 Report

Dear Authors,

please find attached my suggestions

Best Regards

Author Response

Feb 15, 2023.

Dear Editor:

Thank you for the attention you have given to our manuscript originally called “Cobalt-Chromium alloys requires epigenetic machinery to drive the proliferative phenotype of endothelial cells” (jfb-2197692). Indeed, we appreciated the concerns and criticism of the reviewers and the opportunity you have given us to submit a revision of our reworked manuscript.

Importantly, we addressed all the points raised in full, and feel that by incorporating their suggestions our manuscript has been much improved.

Our responses to the raised issues are summarized as follows:

REVIEWER 3:

In this manuscript, the authors developed the possibility of using cobalt-chromium (Co-Cr) based alloys as potential implantable devices. The paper, in general, is very interesting and opens the way to the possibility of using new materials for use in dentistry. Some suggestions should be considered before publication in the Journal of Functional Biomaterials.

Rewriting keywords in alphabetical order.

Reaction. Many thanks for bringing this up. The concern was addressed as requested.

The introduction part can be strengthened with more discussion and references. When the authors discuss the development of new alloys and also new biomaterials, which have significantly improved the mechanical properties of implants, they could use more recent references. (e.g. Materials 2022, 15, 3283, https://doi.org/10.3390/ma15093283; J. Mech. Behav. Biomed. Maters, 125, 2022,104926; Mater. Lett. 306, 2022, 130875, https://doi.org/10.1016/j.matlet.2021.130875; Materials 2022, 15(22), 8208; https://doi.org/10.3390/ma15228208) A comparison with these types of materials might give the reader a better understanding of why it is preferable to use materials such as cobalt-chromium alloys.

Reaction. Many thanks for bringing this up. The introduction section was strengthened by using more recent references.

Rewrite the Materials section as it is a simple list and also quite confusing.

 Reaction. Many thanks for bringing this up. The MM section was now better articulated and hope it attends its demands.

No characterization of these new alloys? E.g. SEM, wettability and surface roughness?

 Reaction. The physicochemical properties of Co-Cr were longer defined and widespread distributed into literature However, the biological effect in response to those materials are barely known. Another point comes from to endothelial cells respond to metallic materials preserving an indirect manner, once the growth of blood vessels respect angiocrine signals. In other words, the endothelial cells do not respond directly interacting to the surface of materials, but in a second step by responding to the molecules released from the materials as well as from blood clot remodeling, fibrin network and degranulation of platelets. A better discussion has been made on this regard.

Line 236: Uniform text style.

 Reaction. Many thanks again, we have made the uniformity of the text style.

Generally improved image resolution.

Reaction. Many thanks again. We have better worked the image resolution. Sorry for some inconvenient.

A separate paragraph should be spent on the conclusions. Here, are too general.

Reaction. Many thanks again, we have reworked the text and a new conclusion was included.

In conclusion, we feel that we have taken into account the concerns of the reviewer in full and that the present version of the manuscript should be acceptable for publication in Journal of Functional Materials.

Looking forward to your evaluation,

WILLIAN FERNANDO ZAMBUZZI

On behalf the authors